# Estimating the potential impact of global research on neglected tropical diseases on population-level indicators of health access, sanitation, and research capacity

David A. Hernandez-Paez[1], Ivan David Lozada-Martinez[1,2,3]*, Juan David Reyes-Duque[4], Sulaiman Kalokoh [5]*

1 Center for Meta-Research and Scientometrics in Biomedical Sciences, Barranquilla, Colombia, 2 Biomedical Scientometrics and Evidence-Based Research Unit, Department of Health Sciences, Universidad de la Costa, Barranquilla, Colombia, 3 Clínica Colsanitas S.A., Clínica Iberoamérica, Barranquilla, Colombia, 4 Facultad de Ciencias para la Salud, Universidad de Manizales, Caldas, Manizales, Colombia, 5 Faculty of Information and Communication Technology, Limkokwing University of Creative Technology, Freetown, Sierra Leone

* ilozada@cuc.edu.co (IDL-M); kalokoh.sulaiman@gmail.com (SK)

## Abstract

### Background

Neglected tropical diseases (NTDs) continue to affect over a billion people, disproportionately impacting low- and middle-income countries (LMICs). While scientific research on NTDs has expanded substantially over the last two decades, it remains unclear whether this growth has translated into measurable improvements in population-level health, development, or research capacity. This study aimed to evaluate the potential impact of global NTDs research on population-level indicators related to health systems, socioeconomic conditions, and research and development.

### Methods

We conducted a longitudinal analysis, integrating bibliometric data from 107,251 NTDs-related publications with country-level indicators from open-access global databases. Countries were stratified by World Bank income classification. Linear regressions, meta-analyses, and meta-regressions were used to examine associations and heterogeneity across indicators and income groups.

### Results

NTDs research output was significantly associated with reduced out-of-pocket health expenditures and current health spending (as % of GDP) in LICs and LMICs, and with expansion of the NTD drug pipeline in LICs and HICs. Strongest and most consistent effects were observed in the WASH domain, particularly reductions in open

**Data availability statement:** The dataset generated and analyzed is available at https://doi.org/10.5281/zenodo.16688508.

**Funding:** The author(s) received no specific funding for this work.

**Competing interests:** The authors have declared that no competing interests exist.

defecation and sanitation-related mortality. However, research capacity gains were concentrated in HICs.

## Conclusions and Implications

While NTDs research has yielded measurable benefits in selected domains and settings, its overall population-level impact remains uneven. Enhancing scientific coherence will require intentional alignment of research agendas with health system needs, equity-driven funding, and stronger translation mechanisms in resource-limited settings.

### Author summary

The global rise in research on neglected tropical diseases, such as Chagas disease, leishmaniasis, and dengue, has raised important questions about its real-world effects. These diseases continue to affect poor communities, especially across Africa, Latin America, and Asia. Over the past decades, thousands of scientific papers have been published, yet uncertainty remains about whether this growing body of work has led to improvements in sanitation, healthcare access, and local scientific development. To explore this, a longitudinal analysis was conducted, linking published research on neglected tropical diseases with population-level indicators related to health access, sanitation, and research capacity. The findings revealed that progress has been uneven. Most research was produced by high-income countries, while improvements in key indicators were slower in low-income countries, where the burden of disease remains greatest. These results suggest that scientific output does not always translate into equitable health benefits. A better alignment between research priorities and the needs of affected populations appears essential to ensure that scientific progress contributes to real-world impact, especially in the most vulnerable settings.

## Introduction

Neglected tropical diseases (NTDs) such as Chagas disease, dengue, and leishmaniasis still affect more than one billion people worldwide, trapping communities in cycles of illness, lost productivity, and poverty [1,2]. According to the World Health Organization (WHO) Global Report on NTDs 2024 [2], 1.62 billion people required at least one NTDs-related intervention in 2022, a figure that, while 26% lower than in 2010, remains far from the 90% reduction targeted for 2030 [2]. The bulk of this need falls on low- and middle-income countries (LMICs), with South-East Asia and sub-Saharan Africa alone accounting for nearly three-quarters of those at risk [2,3].

Over the past two decades, international funding bodies and national governments have invested heavily in NTDs research, under the implicit assumption that more science means better health [4]. New diagnostics, drugs, and implementation strategies

have indeed emerged [5], but exactly how, and how much, this growing body of knowledge translates into measurable improvements in population health remains unknown.

A fundamental blind spot remains: although global health narratives assume that more research means better health [6], no large-scale study has quantified whether the publication of NTDs research over the past two decades has actually translated into measurable improvements in population-level indicators, particularly across different income levels group [7].

Prior impact assessments focus on single diseases or local programs [8], leaving unanswered questions about the global coherence between research effort and real-world impact [2]. This gap fuels a debate: some argue that innovation, wherever it occurs, eventually benefits everyone; others contend that without deliberate alignment of funding and local capacity, research may neglect frontline needs, especially in resource-limited settings [9]. Resolving this controversy is essential for funders, policymakers, and researchers seeking to maximize the real-world value of limited resources [10].

The challenge is scientific coherence, the degree to which research effort aligns with the health needs of different populations [11]. High-income countries (HICs) often produce the bulk of publications [12], while LMICs carry the heaviest NTDs burden [2]. If research truly drives health gains, we would expect countries (or income groups) generating more NTDs science to show faster improvement in NTDs-related health metrics. Yet no study has systematically tested this hypothesis across multiple population health indicators, and economic groups.

To address these gaps, we conducted a longitudinal analysis coupling global NTDs research output with population-level health, socioeconomic, and research indicators, specifically, health system and healthcare access, Water, Sanitation, and Hygiene (WASH), and Research and Development (R&D), stratified by World Bank income groups. Our aim was to quantify the extent to which increased scientific activity on NTDs is associated with favorable trends in the indicators most likely to reflect better population metrics. In this way, the potential impact of scientific research on NTDs was estimated in relation to population-level indicators. By providing this large-scale, data-driven assessment of NTDs research impact across diverse settings, this study offers a practical framework for monitoring scientific coherence and guiding future investment, and also an empirical approach to test the hypothesis that, ideally, more research on NTDs should improve population-level indicators.

## Methods

### Ethical statements

This study was approved by the Scientific Committee of Universidad de la Costa. However, no humans, animals, or medical records were used as units of analysis.

### Study design

Retrospective longitudinal analysis.

### Data sources and collection

First, on March 2, 2025, we performed a comprehensive bibliometric analysis of global NTDs research literature in Scopus, PubMed and Web of Science databases up to 2024. A search strategy was designed using MeSH terms and their equivalents to identify documents that met standard peer-review criteria and focused on the analysis, discussion, investigation, summarization, or evaluation of NTDs (inclusion criteria) [12]. Complete search strategy can be found in S1 File.

To ensure scientific coherence, the terms used were based on the globally prioritized diseases outlined in the WHO Global Report on NTDs 2024 [2]. Accordingly, the following types of publications were excluded: undefined, data paper, retracted, book, erratum, book chapter, and conference paper.

The search was conducted on March 02, 2025, in English. Studies whose original publication was in a language other than English were considered eligible if they included an abstract in this language, fulfilled all inclusion criteria, and did not meet any exclusion criteria. No restrictions were applied regarding the earliest year of publication.

We identified 135,129 NTDs-related articles published between 1904 and 2024. Our analysis included 107,251 articles after filtering to retain only those with identifiable first-author country affiliations (S2 File).

The country of the first author's affiliation was used as a proxy for the paper's origin and categorized according to the World Bank's income classification: HICs, upper-middle-income (UMICs), LMICs, and low-income countries (LICs) [13]. This classification was used to explore potential inequities and differences among groups of countries based on their economic income levels [13]. Additionally, we selected the first author as the primary unit of analysis because, in biomedical sciences, this position typically reflects the leadership in study execution and drafting, thus serving as a robust proxy for the active research capacity driving the agenda

Second, we extracted 75 country-level indicators from official international repositories including the WHO Global Health Observatory [14], World Bank Open Data [15], and Our World in Data (2000–2024) [16]. These indicators covered eight domains: Health System and Healthcare Access; Disease Burden and Infectious Diseases; Demographic and Population Health; Economic and Poverty; Education and Development; Governance and Political; WASH; and R&D. Our analysis primarily focused only on 15 indicators from the Health System, WASH, and R&D domains (S1 Table). The remaining indicators were included as moderators in the meta-regression analysis.

These indicators were extracted annually based on their availability in the consulted databases. This approach ensured a longitudinal alignment between the publication data and global indicators.

## Data standardization

The retrieved publication records from multiple databases were exported in.CSV format, preserving all available metadata, and other relevant attributes. To refine the dataset, an initial independent manual screening was carried out by two researchers. This phase involved removing duplicate entries and assessing titles and abstracts to verify adherence to the predefined inclusion and exclusion criteria. Microsoft Excel 2016 was used for this process. Discrepancies between reviewers were addressed with the involvement of a third evaluator.

## Data synthesis and analysis

All analyses were conducted using R version 4.5.0 [17]. For time-variant indicators, we implemented a hierarchical aggregation approach: first calculating country-level means within each income group and year, then aggregating to income group-level values using publication-weighted means. This methodology ensured that countries with substantial research output appropriately influenced group-level metrics while preventing distortion from outliers with minimal publications [18]. Missing values were handled with explicit removal followed by post-processing conversion of undefined values. Systematic data quality assessment confirmed sufficient information density across all indicators and time periods. This synthesis yielded a merged dataframe integrating bibliometric data (publication counts) and population-level indicators, grouped by year and income level, which served as the primary dataset for analysis.

## Income group-specific regression models

We classified indicators as either dependent or independent variables based on theoretical coherence and interpretability regarding the specific relationships explored in this study, as shown in S1 Table. For dependent indicators, we modeled: Indicator ~ Publication Count. For independent indicators, we modeled: Publication Count ~ Indicator. Separate ordinary least squares regression models were fitted for each indicator-income group combination using annual data from 2000 to 2024, with comprehensive statistics extracted including coefficients ($\beta_1$), standard errors, t-values, p-values, $R^2$ values,

adjusted $R^2$ values, and observation counts. The results yielded $\beta_1$ coefficients, which represent the magnitude of change in the dependent variable per 1-unit increase in the independent variable, accompanied by each model's p-value to indicate statistical significance. Only models with sufficient data points ($n \geq 4$) proceeded to meta-analysis.

### Random-effects meta-analysis

We synthesized evidence across income groups through random-effects meta-analyses using the metagen function (meta package [19]) in R v4.5.0. For each indicator, regression coefficients were pooled with inverse-variance weighting. We employed restricted maximum likelihood (REML) for between-study variance estimation [20], offering improved accuracy over conventional estimators for small study numbers. The Hartung-Knapp-Sidik-Jonkman adjustment corrected for small-sample bias in calculating summary effect sizes with 95% confidence intervals [21].

Heterogeneity was assessed using multiple metrics: $I^2$ (percentage of variance attributable to between-income group variability), $\tau^2$ (absolute between-group variance), Q statistic (weighted sum of squared differences), and heterogeneity p-values. We considered $I^2$ values less than 25% as indicative of low heterogeneity, 25–50% as moderate, 50–75% as substantial, and values greater than 75% as high heterogeneity. Meta-analyses required data from at least two income groups, with comprehensive error handling procedures implemented to address computational challenges. This process generated pooled regression coefficients that maintain the same interpretative framework as the individual coefficients from step one, but represent the aggregated effect across the four income groups, with corresponding 95% confidence intervals providing a measure of precision.

### Meta-regression analysis

To explain heterogeneity in effect sizes across income groups, we conducted meta-regression analyses using all the remaining indicators as potential moderators. Models were implemented through the metafor package [22] in R, following the equation:

$$\theta_i = \beta_0 + \beta_1 X_i + \varepsilon_i m$$

Where $\theta_i$ represents the income group i effect size, $\beta_0$ the intercept, $\beta_1$ the meta-regression coefficient, $X_i$ the moderator value for income group i, and $\varepsilon_i$ the error term encompassing within-group and residual between-group variance.

Meta-regression models required a minimum of four income groups with complete data. For time-varying moderators, we calculated temporal averages across 2000–2024 to derive representative values. Variance estimation employed REML methods with confidence intervals calculated using standard normal-distribution approaches.

For each meta-regression, we calculated pre- and post-moderator heterogeneity metrics, including reduction percentages in $\tau^2$ and $I^2$, residual heterogeneity tests (QE), moderator significance tests (QM), and corresponding p-values. The Benjamini-Hochberg false discovery rate correction addressed multiple comparisons. We report the βmod coefficient indicating the moderator's influence on the original relationship, positive values indicating enhancement and negative values suggesting attenuation, along with statistical significance and resulting $I^2$ values quantifying heterogeneity reduction. Statistical significance was set at a p-value <0.05.

Data processing and statistical code are publicly available at https://doi.org/10.5281/zenodo.16688508, to ensure transparency and reproducibility [23].

## Results

A total of 107,251 articles were retrieved (Fig 1), with original articles being the most frequent type (83.6%), and the United States, India, and Brazil being the top three countries with the most publications (S2 File).

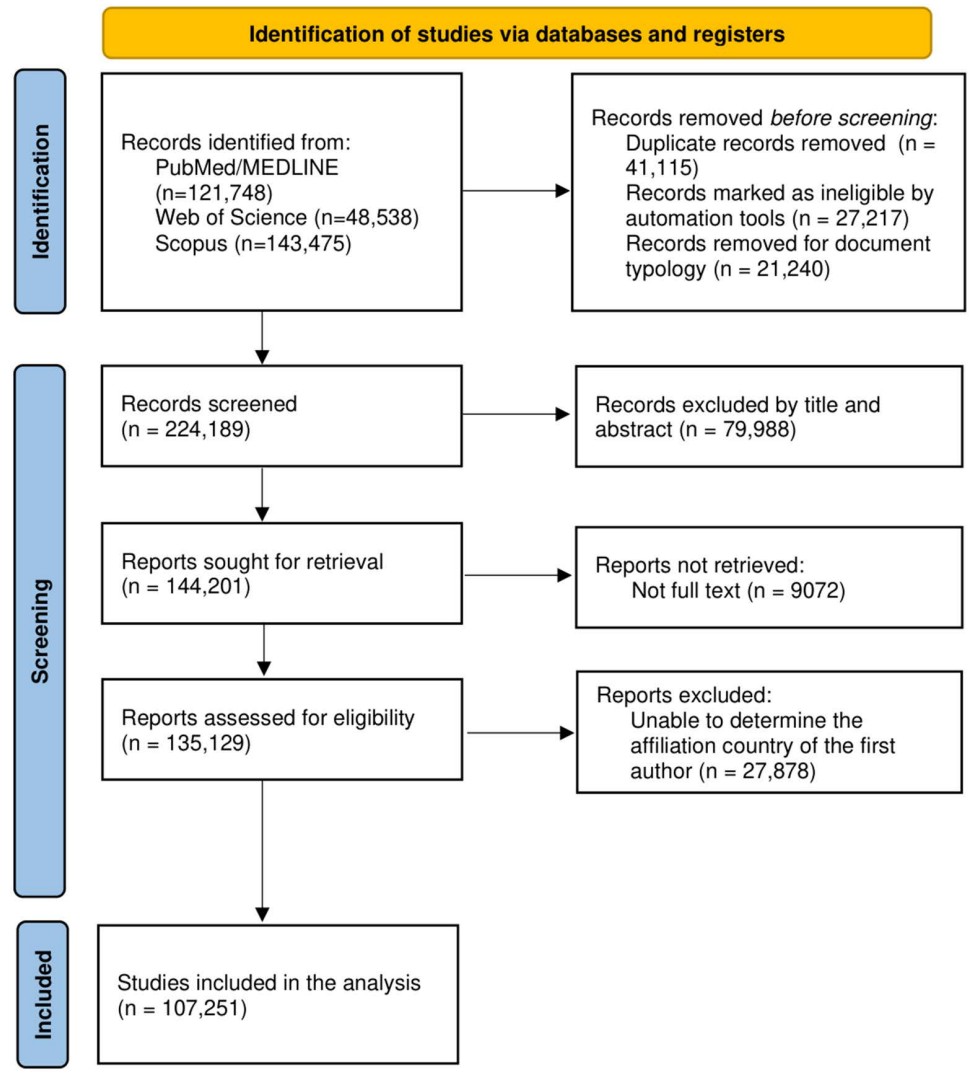

**Fig 1. Flowchart of selected documents.**

### Impact of NTDs research on health system access, R&D, and WASH-specific indicators

NTDs research output demonstrated a significant reduction in the out-of-pocket health expenditure across all income classifications, with the strongest effect observed in LICs ($\beta=-0.001$, $p<0.001$). Furthermore, NTDs research output showed a significant negative association with current health expenditure as percentage of Gross Domestic Product (GDP), with each additional publication corresponding to a reduction of 0.03 percentage points in LICs ($\beta=-0.03$, $p<0.001$) and 0.002 percentage points in LMICs ($\beta=-0.002$, $p<0.01$) (Fig 2 and S3 File).

When evaluating healthcare workforce as the independent variable and research output as the dependent, we identified a significant positive coefficient only in UMICs ($\beta=409.37$, $p<0.05$), representing the number of publications gained per additional physician per 1,000 population. Regarding research output per additional nurse and midwife per 1,000 people, the only significant coefficient was observed in LICs ($\beta=46.22$, $p<0.05$), notably less than the coefficient observed for physicians.

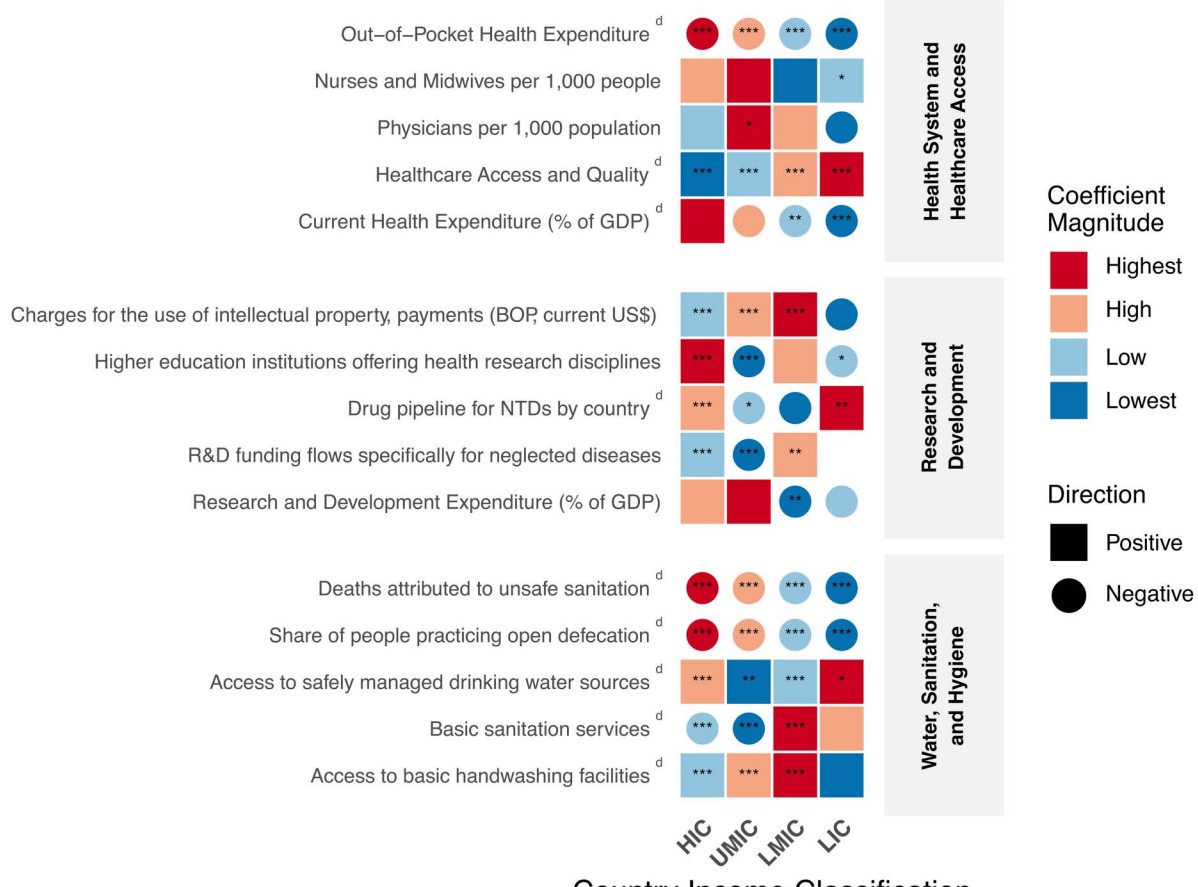

**Fig 2. Effect of biomedical publications on neglected diseases across Health System, Water and Sanitation (WASH), and Research and Development (R&D) indicator domains by World Bank income group classification.** The heatmap displays regression coefficients from linear models examining the relationship between neglected diseases research output and 17 indicators categorized into three domains: 1) Health System and Healthcare Access; 2) R&D; and 3) WASH. Countries are grouped according to World Bank income classifications: HIC (High-Income Countries), UMIC (Upper-Middle Income Countries), LMIC (Lower-Middle Income Countries), and LIC (Low-Income Countries). Color intensity represents coefficient magnitude (red=highest; light orange=high; light blue=low; dark blue=lowest). Shape indicates effect direction (squares=positive associations, circles=negative associations). Statistical significance is denoted by asterisks (* p<0.05, ** p<0.01, *** p<0.001). Blank cells indicate insufficient data for analysis. [d]Used to refer to dependent variables in regression analysis. BOP: Balance of Payments; GDP: Gross Domestic Product.

R&D Expenditure (% of GDP) demonstrated a significant negative association with NTDs publications only in LMICs, where each percentage point increase corresponded to 738.56 fewer publications (β=-738.56, p<0.01), with no significant effects in other income groups. In contrast, targeted R&D funding flows specifically for neglected diseases showed a different pattern: each funding unit increased publication output in HICs (β=0.447, p<0.005) and LMICs (β=7.97, p<0.01), while reducing publications in UMICs (β=-47.03, p<0.001) and showing no significant effect in LICs. NTDs research was also associated with expansion of the drug pipeline for NTDs at the extremes of income distribution, with the strongest effect in LICs (β=0.034, p<0.01), followed by HICs (β=0.001, p<0.001), while yielding inconclusive or negative associations in the middle-income groups (UMICs and LMICs).

The number of higher education institutions offering health research disciplines demonstrated a positive effect on NTDs publication rates exclusively in HICs (β=4956, p<0.001). In contrast, all lower income countries showed either negative associations, as observed in UMICs (β=-2058.41, p<0.001) and LICs (β=-924.1, p<0.05), or no significant effect, as

seen in LMICs. Charges for the use of intellectual property (Balance of Payments [BoP], current US$) exhibited a significant positive association with publication output across all income groups except LICs, where no significant effect was observed. This positive relationship was inversely proportional to income level, with coefficients of $\beta = 2.7$ in HICs, $\beta = 4.67$ in UMICs, and $\beta = 7.98$ in LMICs (all $p < 0.001$).

Each publication on NTDs was significantly associated with reductions in both the share of people practicing open defecation and deaths attributed to unsafe sanitation across all income groups. For open defecation, each additional publication corresponded to greater absolute reductions in LICs ($\beta = -3.01 \times 10^{-3}$, $p < 0.001$) compared to HICs ($\beta = -2.23 \times 10^{-7}$, $p < 0.001$). Similarly, for deaths attributed to unsafe sanitation, the reduction effect was more pronounced in LICs ($\beta = -1.97 \times 10^{-4}$, $p < 0.001$) than in HICs ($\beta = -1.68 \times 10^{-8}$, $p < 0.001$), with both associations following the same pattern across income groups.

NTDs research demonstrated a significant positive association with increased access to basic handwashing facilities in HICs ($\beta = 6.03 \times 10^{-4}$, $p < 0.001$) and LMICs ($\beta = 1.02 \times 10^{-3}$, $p < 0.001$), with the effect magnitude inversely proportional to national income level. Notably, this association was not statistically significant in LMICs. Unexpectedly, each additional NTDs publication was associated with a small but statistically significant reduction in access to basic sanitation services in HICs ($\beta = -1.64 \times 10^{-5}$, $p < 0.001$) and UMICs ($\beta = -7.35 \times 10^{-5}$, $p < 0.001$), while demonstrating a positive impact only in LMICs ($\beta = 1.64 \times 10^{-4}$, $p < 0.001$). In contrast, NTDs research output positively correlated with increased access to safely managed drinking water across all income groups, with the strongest coefficient observed in LMICs ($\beta = 5.15 \times 10^{-4}$, $p < 0.05$).

## Meta-analysis and heterogeneity assessment

Meta-analysis of coefficients across the four income groups for each of the 15 indicators revealed substantial heterogeneity. Notably, the effect of the nurses and midwives (per 1,000 people) as independent variable (Coefficient = 50.45, 95% CI: 21.19–79.72) and the share of the population with access to basic handwashing facilities as dependent variable (Coefficient = 0.00072, 95% CI: 0.00033–0.0011) remained both positive and significant. None of the rest pooled coefficients reached statistical significance based on 95% confidence intervals, and $I^2$ values consistently indicated high heterogeneity (Table 1 and S4 File). This confirmed significant differences in effects across income groups and established the foundation for subsequent meta-regression analyses to identify factors explaining this heterogeneity.

## Determinants of heterogeneity: meta-regression findings

Our meta-regression analyses examined 75 indicators as potential moderators of the interactions shown in the meta-analysis. Table 2 presents only those factors that significantly explained heterogeneity in this relationship, as determined by substantial $I^2$ reduction and statistically significant moderator effects. The comprehensive analytical results are available in S5 File.

Population size significantly moderated the productivity of physicians in generating NTDs research, demonstrating a negative effect where each additional physician produced fewer NTDs publications in more populous countries ($\beta mod = -8.82$, $p < 0.01$), completely eliminating heterogeneity in this relationship ($I^2$ reduced to 0). Conversely, GDP per capita positively moderated physician productivity, with each additional physician per 1,000 people generating more NTDs publications in countries with higher economic development ($\beta mod = 7.8 \times 10^{-7}$, $p < 0.05$, also reducing $I^2$ to 0).

The effect of R&D expenditure (% of GDP) on NTDs article production was significantly diminished by the level of extreme poverty ($\beta mod = -3.29 \times 10^{-8}$, $p < 0.01$, reducing $I^2$ to 0). Regarding access to basic handwashing facilities, NTDs research demonstrated a significantly stronger positive impact in countries with larger populations ($\beta mod = 6.5 \times 10^{-15}$, $p < 0.01$, reducing $I^2$ to 0) and in countries with higher poverty levels ($\beta mod = 1.98 \times 10^{-14}$, $p < 0.01$, reducing $I^2$ to 0), compared to less populated and wealthier nations.

**Table 1. Meta-analysis of the relationship between neglected diseases research output and health system access, research and development, and water and sanitation indicators across world bank income groups.**

| Indicator | Coefficient | [95% CI] | $I^2$ | Q |
|---|---|---|---|---|
| **Health System and Healthcare Access** | | | | |
| Current Health Expenditure (% of GDP) | -0.007 | [-0.03 – 0.01] | 0.72 | 11.09* |
| Healthcare Access and Quality | 0.057 | [-0.072 – 0.18] | 0.98 | 179.31* |
| Nurses and Midwives (per 1,000 people) | 50.45 | [21.19 – 79.72] | 0 | 0.77 |
| Out-of-Pocket Expenditure on Health | -0.0003 | [-0.001 – 0.0004] | 0.98 | 179.2* |
| Physicians (per 1,000 people) | 155.60 | [-208.61 – 519.83] | 0.59 | 7.40 |
| **Research and Development (R&D)** | | | | |
| Charges for the use of intellectual property, payments (BoP, current US$) | $2.62 \times 10^{-7}$ | [$-1.03 \times 10^{-6}$ – $1.56 \times 10^{-6}$] | 0.97 | 141.56* |
| Distribution of R&D funding flows for neglected diseases by country | -12.23 | [-85.59 – 61.13] | 0.96 | 54.95* |
| Drugs pipeline for neglected tropical diseases by country | 0.0056 | [-0.020 – 0.031] | 0.91 | 35.78* |
| Higher education institutions offering disciplines related to research for health in 2023 | 956.93 | [-4184.29 – 6098.16] | 0.96 | 85.78* |
| Research and Development Expenditure (% of GDP) | -198.53 | [-765.36 – 386.3] | 0.69 | 9.76* |
| **Water, Sanitation and Hygiene (WASH)** | | | | |
| Share of deaths attributed to unsafe sanitation | -0.000061 | [-0.0002 – 0.00008] | 0.99 | 664.02* |
| Share of people practicing open defecation | -0.00085 | [-0.003 – 0.0013] | 0.99 | 847.8* |
| Share of the population using basic sanitation service | 0.000022 | [-0.00014 – 0.00019] | 0.99 | 593.47* |
| Share of the population using safely managed drinking water sources | 0.00017 | [-0.00008 – 0.00043] | 0.96 | 87.17* |
| Share of the population with access to basic handwashing facilities | 0.00072 | [0.00033 – 0.0011] | 0.78 | 13.75* |

BoP: Balance of Payments; GDP: Gross Domestic Product; R&D: Research and Development. Asterisks denote $p < 0.05$ for Q statistic.

**Table 2. Socioeconomic and demographic moderators explaining heterogeneity in health system access, research and development, and water and sanitation indicators across world bank income groups: meta-regression analysis with heterogeneity reduction.**

| Moderator | $\beta_{meta}$ [95% CI] | $I^2$ (no mod) | $I^2$ (w/ mod) | $\beta_{mod}$ | p |
|---|---|---|---|---|---|
| **Physicians (per 1,000 people)** | | | | | |
| Population in year | 155 [-80 – 391.3] | 0.59 | 0 | -8.82 | $p < 0.01$ |
| GDP per capita | 155 [-80 – 391.3] | 0.59 | 0 | $7.8 \times 10^{-7}$ | $p < 0.05$ |
| **Research and Development Expenditure (% of GDP)** | | | | | |
| Number of people living in extreme poverty | -189.53 [-535.86 – 156.79] | 0.69 | 0 | $-3.29 \times 10^{-8}$ | $p < 0.01$ |
| **Share of population with access to basic handwashing facilities** | | | | | |
| Population in year | 0.00072 [0.00049 – 0.00095] | 0.78 | 0 | $6.5 \times 10^{-15}$ | $p < 0.01$ |
| Number of people living in extreme poverty | 0.00072 [0.00049 – 0.00095] | 0.78 | 0 | $1.98 \times 10^{-14}$ | $p < 0.01$ |

GDP: Gross Domestic Product.

The association between publications and R&D expenditure (% of GDP) was moderated by the number of people living in extreme poverty ($\beta mod = -3.29 \times 10^{-8}$, $p < 0.01$, reducing $I^2$ to 0). This indicates that poverty levels significantly influence how scientific publications translate to research investment. For WASH indicators, the impact of publications on population access to basic handwashing facilities was significantly moderated by both population size ($\beta mod = 6.5 \times 10^{-15}$, $p < 0.01$) and extreme poverty levels ($\beta mod = 1.98 \times 10^{-14}$, $p < 0.01$), both reducing heterogeneity from $I^2 = 0.78$ to $I^2 = 0$.

## Discussion

This study sought to determine whether the growing body of scientific research on NTDs has translated into measurable improvements across population-level indicators, particularly in low income regions where disease burden is most concentrated. By analysing NTDs-related publications against specific indicators across population health, socioeconomic conditions, and research capacity, we found evidence of potentially partial scientific coherence, with important differences in strength, direction, and equity of associations depending on the income level of countries.

Within the health system and access domain, multiple indicators showed statistically significant potential associations, but with varying implications. Notably, research output was potentially associated with a reduction in out-of-pocket health expenditure, particularly in LICs, where each additional NTDs publication estimated a significant decrease (β = -0.001, p < 0.001). Similarly, a negative possible association with current health expenditure as a percentage of GDP was observed in both LICs (p < 0.001) and LMICs (p < 0.01), suggesting that NTDs research may be indirectly contributing to more efficient healthcare financing models in settings with weaker systems [24].

However, not all effects were uniformly distributed. The availability of physicians per 1,000 population was positively associated with an estimated increase in publication output only in UMICs, while the number of nurses and midwives showed a significant potential relationship in LICs. These patterns suggest that the translation of health workforce expansion into research productivity could be both context-specific and asymmetrical [25], depending on local systems and how health human resources are integrated into scientific agendas [26].

In the R&D domain, findings were more complex. Surprisingly, increased national R&D expenditure (% of GDP) was associated with an estimated reduction in NTDs-related publications in LMICs (β = -738.56, p < 0.01), possibly reflecting a shift in investment priorities away from NTDs, or inefficiencies in funding allocation [27]. In contrast, targeted R&D funding flows for NTDs were associated with potentially increased publication output in HICs (p < 0.005) and LMICs (p < 0.01), but had a negative impact in UMICs and no measurable effect in LICs. This divergence reinforces a key interpretation of this study. Global R&D ecosystems possibly tend to amplify the capacity of countries that already have baseline infrastructure, while failing to correct structural inequities in more fragile contexts [28].

Notably, research output was significantly linked to the growth of the drug development pipeline for NTDs in both LICs (p < 0.01) and HICs (p < 0.001), but not in middle-income countries. This bimodal pattern may reflect two opposing dynamics. In LICs, donor-funded pipelines respond to urgent unmet needs, while in HICs, innovation systems respond to market incentives or academic productivity [29]. In middle-income countries, research may be abundant, but translation into product development is constrained by weak public-private partnerships [30].

The most consistent and policy-relevant signals of research impact emerged in the WASH domain. Publications on NTDs were significantly associated with potential reductions in the share of the population practicing open defecation and with deaths attributable to unsafe sanitation across all income groups, with the strongest effects consistently observed in LICs (both p < 0.001). This finding is perhaps the clearest signal of scientific coherence in this analysis. Where research is aligned with the environmental determinants of NTDs, such as hygiene and sanitation, real-world benefits appear to follow, especially in the most vulnerable countries [31].

Access to basic handwashing facilities also showed a positive association in HICs and LMICs, with the pooled meta-analysis proposing this indicator as one of the few with consistent statistical significance across groups (β = 0.00072, 95% CI: 0.00033–0.0011). However, a surprising and concerning pattern was observed with access to basic sanitation services, where each NTDs publication was associated with a small but significant reduction in access in HICs and UMICs, while only LMICs showed a positive trend. This may reflect reverse causality, with NTDs research expanding in response to sanitation crises rather than driving improvements, or it may indicate deeper systemic inefficiencies in translating evidence into public infrastructure investment in high-income contexts [32].

Meta-regression results further proposed that these relationships were strongly moderated by poverty levels and population size. For example, the estimated effect of research on handwashing access was more pronounced in more

populous and poorer countries, suggesting that research can translate into impact when aligned with scale and need [33]. Conversely, the relationship between R&D expenditure and publication volume was significantly dampened by extreme poverty, suggesting that structural deprivation impairs the return on scientific investment [34].

Altogether, these findings partially support the presence of scientific coherence, especially in the WASH domain and in certain health financing indicators. However, they also propose critical weaknesses in coherence across the broader health and R&D systems, particularly in LMICs and LICs, where research often fails to convert into sustained improvements in health infrastructure or innovation capacity [35]. The implication is that NTDs research may be not inherently transformative unless it is embedded in contexts that allow for uptake, implementation, and equitable benefit distribution [5,6].

This analysis challenges simplistic narratives that equate publication volume with progress. It shows that while scientific production on NTDs has increased dramatically over the past two decades, its measurable impact remains fragmented, often inequitable, and in some domains, statistically negligible [2]. The strongest observed effects point to domains where public health infrastructure and behaviour intersect with research, suggesting that integration between scientific and operational systems is essential [2,4–6].

### Policy translation

Going forward, global health funders, research institutions, and policy actors must recognize that equity in knowledge production does not guarantee equity in outcomes [36,37]. Strengthening research systems in LICs, ensuring the use of evidence for service delivery, and designing feedback mechanisms between knowledge and policy will be critical if NTDs research is to fulfil its intended public health purpose [10].

These findings falsify the conjecture that knowledge accumulation automatically drives progress. Instead, science requires development coherence (defined as the alignment where higher levels of structural development support research capacity and translation) to be effective. Consequently, funding agencies and the WHO Road Map must move beyond publication metrics and actively finance the translation mechanisms that bridge the gap between bibliometric output and population health. Key actionable recommendations include prioritizing implementation studies to operationalize findings, strengthening local research capacity to ensure uptake, and promoting South-South collaborations to foster context-specific solutions. Without this intentional alignment, the scientific enterprise risks becoming self-referential and disconnected from the structural realities of the populations it aims to serve.

### Limitations

Among the strengths of this study is its integrative approach, combining scientometrics and epidemiological data with socioeconomic, health and infrastructural indicators. This multi-dimensional lens provides a more nuanced picture of research impact than publication counts or citations alone. Additionally, the use of longitudinal meta-regression allows for trend interpretation over time, offering insights into the temporal dynamics between research and development outcomes.

However, limitations must be acknowledged. First, causality cannot be inferred due to the observational design. Second, reliance on indexed publications may overlook local grey literature and non-English outputs, potentially introducing publication bias favoring high-income countries and underestimating research activity in LMICs. Fourth, the analysis of human resources was limited to physicians and nurses, potentially overlooking the contributions of non-physician researchers, such as those in veterinary medicine and environmental health. Finally, the complexity of social determinants of health means that NTDs-related outcomes are influenced by numerous non-research factors (e.g., governance, conflict, climate), which are difficult to isolate.

### Conclusions

This longitudinal analysis proposes that while scientific research on NTDs has expanded significantly over the last decades, its translation into measurable improvements in population-level indicators remains limited, context-dependent,

and unevenly distributed across income groups. The hypothesis that higher volumes of NTDs research are associated with systematic gains in public health, socioeconomic conditions, and research capacity was only partially supported.

Significant associations were found primarily in the WASH domain, particularly reductions in open defecation and sanitation-related mortality, and to a lesser extent in indicators related to health financing and scientific productivity, such as out-of-pocket expenditure and domestic R&D output. These effects were strongest in LMICs, indicating that when research is closely aligned with urgent public health needs and foundational infrastructure, it has the potential to produce tangible population-level benefits.

However, across several core indicators, especially those reflecting disease burden, healthcare access, and national innovation systems, no consistent or statistically significant associations were observed, particularly in the countries with the highest disease burden. This finding suggests that the global distribution of NTDs research does not consistently align with population needs, nor does it ensure impact in the absence of systemic implementation and policy integration.

In light of these findings, we argue that scientific coherence must be reframed not as the volume of research produced, but as the ability of that research to generate equitable, sustainable improvements in the conditions that sustain NTDs. Future strategies must prioritize local research capacity, invest in translation mechanisms, and reorient evaluation frameworks toward outcomes that reflect real-world health and development gains, particularly in the settings where the burden is greatest. Finally, the methodological framework introduced here, with its open-access code, is adaptable to other domains; future applications should extend this analysis to policy output indicators, such as national legislation and domestic funding, to provide a more comprehensive view of research impact.

## Supporting information

**S1 File.** Search strategy in databases.
(DOCX)

**S2 File.** Total publication count by country, income level, and study design.
(XLSX)

**S3 File.** Linear regression models characteristics and full results.
(CSV)

**S4 File.** Meta-analysis characteristics and full results.
(CSV)

**S5 File.** Meta-regression full results.
(CSV)

**S1 Table.** Indicators extracted for the analyses.
(DOCX)

## Author contributions

**Conceptualization:** David A. Hernandez-Paez, Ivan David Lozada-Martinez, Juan David Reyes-Duque, Sulaiman Kalokoh.

**Data curation:** David A. Hernandez-Paez, Ivan David Lozada-Martinez, Juan David Reyes-Duque.

**Formal analysis:** David A. Hernandez-Paez, Ivan David Lozada-Martinez, Juan David Reyes-Duque, Sulaiman Kalokoh.

**Investigation:** David A. Hernandez-Paez, Ivan David Lozada-Martinez, Juan David Reyes-Duque, Sulaiman Kalokoh.

**Methodology:** Ivan David Lozada-Martinez.

**Writing – original draft:** David A. Hernandez-Paez, Ivan David Lozada-Martinez, Juan David Reyes-Duque, Sulaiman Kalokoh.

**Writing – review & editing:** David A. Hernandez-Paez, Ivan David Lozada-Martinez, Juan David Reyes-Duque, Sulaiman Kalokoh.

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
