## [Decision Letter · Decision Letter 0]

23 Jan 2026

PNTD-D-25-01315

Measuring the Potential Impact of Global Research on Neglected Tropical Diseases on Population-Level Indicators of Health Access, Sanitation, and Research Capacity

Dear Dr. Kalokoh,

Thank you for submitting your manuscript to PLOS Neglected Tropical Diseases. After careful consideration, we feel that it has merit but does not fully meet PLOS Neglected Tropical Diseases's publication criteria as it currently stands. Therefore, we invite you to submit a revised version of the manuscript that addresses the points raised during the review process.

Please submit your revised manuscript within by Mar 24 2026 11:59PM. If you will need more time than this to complete your revisions, please reply to this message or contact the journal office at plosntds@plos.org. Please include the following items when submitting your revised manuscript:

We look forward to receiving your revised manuscript.

Kind regards,

Stephanie N. Seifert, Ph.D.

Section Editor

Dileepa Ediriweera

Section Editor

Shaden Kamhawi

co-Editor-in-Chief

Paul Brindley

co-Editor-in-Chief

**Additional Editor Comments:**

The reviews for this manuscript are generally positive and highlight the impressive scope and need for this study. Reviewers have brought up a few issues to be addressed prior to publication. We hope that you will consider revising the manuscript following reviewers comments.

**Journal Requirements:**

Please upload all main figures as separate Figure files in .tif or .eps format. For more information about how to convert and format your figure files please see our guidelines:

**Reviewers' Comments:**

Reviewer's Responses to Questions

**Key Review Criteria Required for Acceptance?**

**Methods**

-Are the objectives of the study clearly articulated with a clear testable hypothesis stated?

-Is the study design appropriate to address the stated objectives?

-Is the population clearly described and appropriate for the hypothesis being tested?

-Is the sample size sufficient to ensure adequate power to address the hypothesis being tested?

-Were correct statistical analysis used to support conclusions?

-Are there concerns about ethical or regulatory requirements being met?

Reviewer #1: -

Reviewer #2: (No Response)

Reviewer #3: I suggest that hypotheses and expectations are more clearly described in the introduction. For example what do you expect for each income group and for each indicator? At this time, it feels like an overwhelming set of indicators being used with no clear justification as to why each specific one needs to be included.

There’s a large difference between the output from a narrative vs systematic review. I’m not convinced you should have them put in the same category. I suggest you also explicitly describe what was done to harmonize variables because the harmonization processes may not be agreeable to all readers.

It will be helpful to have a table that clearly labels exclusion and inclusion criteria. Alignment with these criteria keeps coming up in sections of the paper and it’s helpful if the reader has a place to go where all of these are clearly listed.

More justification is needed as to why some variables were categorized as response vs predictor vs moderator. Perhaps a flow chart or some method because right now it seems very subjective.

Focusing on just physicians and nurses is ignoring the non-physicians (environmental health and veterinary medicine) who also conduct research in these types of fields so I think the picture may be incomplete here. At least mention this as a limitation in the discussion.

Can you elaborate on the definition of retrospective longitudinal analysis? I would categorize this study as a systematic literature review with a meta-analysis.

Provide the dates of database search explicitly. So studies of any type whether empirical or theoretical or editorial were included? I realize you mention something along these lines later but more clarification is needed on the nature of studies pursued (what study designs did you consider).

In the R code, there is annotation about low to high heterogeneity, which will be useful to have in the main text to understand your definitions.

**Results**

-Does the analysis presented match the analysis plan?

-Are the results clearly and completely presented?

-Are the figures (Tables, Images) of sufficient quality for clarity?

Reviewer #1: -

Reviewer #2: (No Response)

Reviewer #3: The results generally match the methods. I appreciate the supplemental files where I was able to see all the results.

Please provide a summary of studies found - what you included and excluded and why and years/continents represented. It would be good to include countries where most pubs came from and the most common NTDs researched.

Please also give a summary as to whether assumptions of using regression have been met.

Table 1: Please define how variable type was assigned. I suggest the table moves to supplement and include in a new Table 1 only the indicators that were used as non-moderators.

Table 2: Why not p-values next to Q column here?

Figure 1. Can you briefly state why were reports not retrieved? Just to ensure this didn’t add too much bias in the results. I suggest this figure moves to supplement. If it stays, it should go under results (not in methods).

Figure 2: Suggest switching the right column headings to vertical headings over each set of indicators so you can expand the heat map portion. Perhaps think about suitable ways of making some of those long indicators into shorter ones because it’s hard to read the heat map otherwise. Generally speaking being able to look at point estimates for each income group across each indicator would provide more information and align with the text better. Not sure why there's a white space for LIC R&D funding flows.

Can you provide a bar chart of number of pubs per year or something along those lines to give the reader insight as to the temporal trend?

It would also be useful if you had a choropleth global map showing where pubs are coming from.

**Conclusions**

-Are the conclusions supported by the data presented?

-Are the limitations of analysis clearly described?

-Do the authors discuss how these data can be helpful to advance our understanding of the topic under study?

-Is public health relevance addressed?

Reviewer #1: in the Discussion section:

Expand the discussion on practical and policy implications, specifying how the findings can inform funding agencies, WHO initiatives, or national NTD programs.

Clarify that negative or non-significant results (e.g., absence of association with DALYs) do not indicate “no impact,” but rather a lack of statistical traceability. This distinction will help avoid possible misinterpretations.

Consider adding a final subsection on “policy translation”, including actionable recommendations such as strengthening local research capacity, promoting South–South collaborations, and encouraging implementation studies.

in the Limitation section:

Add a note about potential publication bias, since the most visible studies tend to come from high-income countries.

Specify that the use of annual, country-level averages implies a lack of temporal and spatial granularity, which could dilute local or short-term effects.

Suggest the future inclusion of policy output indicators (e.g., national health legislation or domestic funding for NTD research) to provide a more comprehensive view of research impact.

Reviewer #2: (No Response)

Reviewer #3: I would suggest removing p and beta values from the discussion. You may want to use language that’s less statistical here. I think the most compelling results are WASH-related and should come up sooner.

How would the limitations impact your conclusions however? I think the limitations you mentioned especially second and third are very important to consider in how your results could be affected and whether your conclusions would be changed. Consider how publication biases and not testing for interactions (e.g. poverty and population) may also impact your conclusions.

Generally, I recommend you avoid causative language because as you mentioned the study is observational. So in some areas where there are strong recommendations for funding allocation etc try to tone that down reminding the reader that there are caveats and other variables could be playing a role here that were not measured.

**Editorial and Data Presentation Modifications?**

Reviewer #1: -

Reviewer #2: (No Response)

Reviewer #3: Line 123: Replace sanity with sanitation.

Line 438: why as expected? This will be clearer if you are more explicit with your expectations in the intro.

Line 540: Replace larger with more populous to avoid thinking of land mass.

Line 561: Change to fulfill.

Please provide R package citations – the Zenodo reference lists more R packages that were used than what is shown in the manuscript.

**Summary and General Comments**

Reviewer #1: This is a solid, courageous, and necessary study that opens a promising line of research — assessing not only how much science is produced, but how much it truly contributes to global health. However, it represents only a first step; mixed-method approaches (quantitative + qualitative) are recommended to better understand the mechanisms of research translation and implementation.

I recommend few minor changes and then, in my opinion, the manuscript can be published

Reviewer #2: This manuscript presents an ambitious longitudinal analysis linking NTD research output to population-level health, socioeconomic, and research capacity indicators across income groups. While the study addresses an important gap in understanding research impact, several methodological and interpretive issues need attention. The paper needs recalibration to match conclusions to what the observational design can support.

1.The title, abstract, and throughout claim to measure ‘‘impact’’ and ‘‘effects’’, but the methods only establish cross-sectional correlations. The authors should consider whether this is fundamentally a study showing research-outcome correlations vary by setting (more defensible) versus research impact on outcomes (much harder to establish with this design).

2.The authors employ random-effects meta-analysis to synthesize regression results across income groups, which is unconventional since meta-analysis is typically designed to combine findings from independent studies rather than strata within a single dataset. While this approach is not methodologically incorrect, it requires clear justification for why meta-analysis was chosen over more standard alternatives such as mixed-effects models with income group as a random effect, stratified regression with interaction terms, or formal heterogeneity tests like the Chow test. Without this justification, readers cannot assess whether the chosen analytical framework appropriately addresses the research question or whether simpler, more transparent methods might be preferable.

3.The authors correlate research publications with health indicators from the same year, which cannot demonstrate causality since research typically requires several years to translate into population health improvements. This approach only shows that countries producing more research also have better health in the same time period, but cannot determine whether research causes better health or whether healthier, wealthier countries simply produce more research. The authors must either introduce temporal lags (comparing past research with future health outcomes) or explicitly reframe their findings as correlational patterns rather than evidence of research impact.

4.Attributing research to the first author’s country is problematic because it miscounts where research actually benefits populations. For example, a US-led study on Kenyan diseases gets credited to the US, not Kenya. This systematically inflates high-income countries’ apparent contributions since they dominate first authorship even for work conducted in low-income settings. The authors’ own finding that research capacity grows in rich countries while health improvements occur in poor countries suggests this attribution method fails to track where research impact actually occurs.

5.The Results are clearly presented statistically, but the interpretation throughout overstates what the data can show. The fundamental issue—that same-year correlations cannot demonstrate research impact—undermines the core narrative.

6.The strongest findings are in WASH indicators, but many WASH improvements have occurred through infrastructure investment independent of biomedical research. The mechanism linking NTD publications to handwashing facilities remains unclear. This domain may reflect confounding by general development more than others. The authors should discuss alternative explanations for WASH improvements, clarify plausible mechanistic pathways between publications and infrastructure outcomes, and consider whether WASH indicators are appropriate measures of research impact.

7.Did WASH improvements accelerate after NTD research increased, or did both trend upward independently? Without temporal sequencing, the authors just showing correlation.

8.Current conclusion overstates what the data can support

9.The manuscript’s language is generally strong, but minor grammatical corrections (e.g., sanity to sanitation, line 132), tense consistency, and improved precision in causal wording are required for professional polish and journal compliance.

Reviewer #3: The authors should be commended for undertaking such an extensive study. I appreciate all the work that was done and the thought that went into putting this study together. I also appreciate how they provided the data and the R code. I do have some significant concerns however that may be able to be addressed through providing the reader with more justification in the methods and intro although some additional data analyses and visualizations will be necessary.

The path from pubs (research output) to policy/regulations that impact population health outcomes is more complicated than what is implied here and there are likely significant time lags from research to population health outcomes that have not been considered. The assumption that research predominantly drives health gains and that countries with more NTD pubs will have better health metrics needs to be fleshed out more to be convincing.

Although I appreciate the complicated statistical methods, I am not convinced of the conceptual approach here. The authors state that HIC countries already produce most of the NTD research but what about the ones doing so in collaboration with lower income groups to address their problems? The authors assign country origin based on where the first author of the paper is from but I’m not sure this accounts for co-authors that could be from lower income countries (and why not senior author for example). I will need stronger justification in methods and intro.

Furthermore, I’m not convinced that aggregating all NTDs together (from parasitic to viral diseases) and looking at weighted pub numbers as a response or predictor variable indicative of research output is a suitable approach. Some NTDs receive more attention than others, and some are more prevalent in one income group than the other so I am concerned as to whether a lot of important signals are being muddled because NTDs are aggregated. For example what if there are clear signals for malaria but not for snake bites and by aggregating everything together you are mixing signals? I think for me to be convinced of this approach I will need stronger justification in the methods and intro as to why this approach is suitable.

I'm also unclear as to why the conclusions may not be a mere reflection of multiple factors beyond pubs where over time the global society has improved public health measures. For example, factors other than number of pubs (e.g. word of mouth etc) may have improved sanitation and hygiene over time and positive associations indicate that in more recent years we have more pubs but we also have more community awareness (independent of pubs) that happens to correlate with more hygiene and sanitation. If this was addressed in the paper I was not able to catch it and thus may need more explicit explanation. Because the authors state that NTD research has expanded over the last two decades, could they look how the population health metrics have changed before and after this large expansion in literature? Unless the expansion is less drastic than that. Because otherwise, why go back to the literature from the early 1900s?

PLOS authors have the option to publish the peer review history of their article (what does this mean?). If published, this will include your full peer review and any attached files.

Reviewer #1: No

Reviewer #2: **Yes:** Dai Kuang

Reviewer #3: No

**Figure resubmission:** While revising your submission, we strongly recommend that you use PLOS’s NAAS tool (https://ngplosjournals.pagemajik.ai/artanalysis) to test your figure files. NAAS can convert your figure files to the TIFF file type and meet basic requirements (such as print size, resolution), or provide you with a report on issues that do not meet our requirements and that NAAS cannot fix.
---

## [Decision Letter · Decision Letter 1]

6 May 2026

Dear Mr Kalokoh,

We are pleased to inform you that your manuscript 'Measuring the Potential Impact of Global Research on Neglected Tropical Diseases on Population-Level Indicators of Health Access, Sanitation, and Research Capacity' has been provisionally accepted for publication in PLOS Neglected Tropical Diseases.

Best regards,

Dileepa Ediriweera

Section Editor

Dileepa Ediriweera

Section Editor

Shaden Kamhawi

co-Editor-in-Chief

Paul Brindley

co-Editor-in-Chief

Reviewer's Responses to Questions

**Key Review Criteria Required for Acceptance?**

**Methods**

-Are the objectives of the study clearly articulated with a clear testable hypothesis stated?

-Is the study design appropriate to address the stated objectives?

-Is the population clearly described and appropriate for the hypothesis being tested?

-Is the sample size sufficient to ensure adequate power to address the hypothesis being tested?

-Were correct statistical analysis used to support conclusions?

-Are there concerns about ethical or regulatory requirements being met?

Reviewer #2: (No Response)

**Results**

-Does the analysis presented match the analysis plan?

-Are the results clearly and completely presented?

-Are the figures (Tables, Images) of sufficient quality for clarity?

Reviewer #2: (No Response)

**Conclusions**

-Are the conclusions supported by the data presented?

-Are the limitations of analysis clearly described?

-Do the authors discuss how these data can be helpful to advance our understanding of the topic under study?

-Is public health relevance addressed?

Reviewer #2: (No Response)

**Editorial and Data Presentation Modifications?**

Reviewer #2: (No Response)

**Summary and General Comments**

Reviewer #2: The authors have carefully addressed all of my comments and revised the manuscript accordingly. The revisions have improved the clarity and quality of the paper. I have no further concerns and recommend the manuscript for publication.

PLOS authors have the option to publish the peer review history of their article (what does this mean?). If published, this will include your full peer review and any attached files.

Reviewer #2: No

---

## [Editor Report · Acceptance letter]

Dear Mr Kalokoh,

We are delighted to inform you that your manuscript, "Estimating the Potential Impact of Global Research on Neglected Tropical Diseases on Population-Level Indicators of Health Access, Sanitation, and Research Capacity," has been formally accepted for publication in PLOS Neglected Tropical Diseases.

Best regards,

Shaden Kamhawi

co-Editor-in-Chief

Paul Brindley

co-Editor-in-Chief
